# A Universal Fluorescent Immunochromatography Assay Based on Quantum Dot Nanoparticles for the Rapid Detection of Specific Antibodies against SARS-CoV-2 Nucleocapsid Protein

**DOI:** 10.3390/ijms23116225

**Published:** 2022-06-02

**Authors:** Zehui Li, Aiping Wang, Jingming Zhou, Yumei Chen, Hongliang Liu, Yankai Liu, Ying Zhang, Peiyang Ding, Xifang Zhu, Chao Liang, Yanhua Qi, Enping Liu, Gaiping Zhang

**Affiliations:** 1School of Life Science, Zhengzhou University, Zhengzhou 450001, China; lizehui2091@163.com (Z.L.); pingaw@126.com (A.W.); zhjingming@126.com (J.Z.); yumeichen2012@163.com (Y.C.); liuhongliang55@126.com (H.L.); lyk@zzu.edu.cn (Y.L.); zhangy2021@zzu.edu.cn (Y.Z.); dingpeiyang1990@163.com (P.D.); xifang11@hotmail.com (X.Z.); chaonvlc@163.com (C.L.); yanhuaqi2007@163.com (Y.Q.); liuenping@ioz.ac.cn (E.L.); 2School of Advanced Agriculture Sciences, Peking University, Beijing 100871, China; 3Longhu Laboratory of Advanced Immunology, Zhengzhou 450000, China

**Keywords:** SARS-CoV-2, fluorescent immunochromatography assay, nucleocapsid protein, quantum dots, point-of-care detection

## Abstract

Severe Acute Respiratory Syndrome Coronavirus 2 (SARS-CoV-2) is the pathogenic agent leading to COVID-19. Due to high speed of transmission and mutation rates, universal diagnosis and appropriate prevention are still urgently needed. The nucleocapsid protein of SARS-CoV-2 is considered more conserved than spike proteins and is abundant during the virus’ life cycle, making it suitable for diagnostic applications. Here, we designed and developed a fluorescent immunochromatography assay (FICA) for the rapid detection of SARS-CoV-2-specific antibodies using ZnCdSe/ZnS QDs-conjugated nucleocapsid (N) proteins as probes. The nucleocapsid protein was expressed in *E.coli* and purified via Ni-NTA affinity chromatography with considerable concentration (0.762 mg/mL) and a purity of more than 90%, which could bind to specific antibodies and the complex could be captured by *Staphylococcal* protein A (SPA) with fluorescence displayed. After the optimization of coupling and detecting conditions, the limit of detection was determined to be 1:1.024 × 10^5^ with an IgG concentration of 48.84 ng/mL with good specificity shown to antibodies against other zoonotic coronaviruses and respiratory infection-related viruses (*n* = 5). The universal fluorescent immunochromatography assay simplified operation processes in one step, which could be used for the point of care detection of SARS-CoV-2-specific antibodies. Moreover, it was also considered as an efficient tool for the serological screening of potential susceptible animals and for monitoring the expansion of virus host ranges.

## 1. Introduction

Coronavirus disease (COVID-19) was firstly reported due to unknown pneumonia reported in Wuhan, China, inducing a large-scale epidemic worldwide with 520 million people infected and 6.3 million deaths confirmed by now [1]. It was found to be caused by the novel coronavirus, Severe Acute Respiratory Syndrome Coronavirus 2 (SARS-CoV-2) [2].

SARS-CoV-2 is a virus with a single-strand positive sense RNA, classified as a member of *betacoronavirus* genus of the *Coronaviridae* family [3]. A full-length reference sequence (Genbank ID: NC_045512) was obtained at the early stage of the outbreak, which indicated 79.5% identity relative to SARS-CoV and 96% identity relative to bat coronavirus RaTG13 [4]. The genome encodes four structural proteins including spike (S), envelop (E), membrane (M), nucleocapsid (N) and several non-structural proteins and accessory proteins [5]. N protein is a multifunctional protein [6] that is in charge of RNA-binding and packing them into helical nucleocapsid structure or ribonucleoprotein (RNP) complex; it plays important roles in providing nuclear-import signals, interfering cell processes, virus replication and RNA package [7,8]. Quantitative measurements of clinical antibody samples against nucleocapsid and spike proteins were analyzed, reporting that antibodies relative to N proteins are more sensitive (100%) than those against S proteins (91%) for detecting early infection [9,10]. Wang et al. [11] provided a lateral flow kit based on a selenium nanoparticle (SeNP)-modified N protein for the simultaneous detection of anti-SARS-CoV-2 IgG and IgM in human serum with a sensitivity of 93.33% and specificity of 97.34%. Cavalera et al. [12] reported a multi-line lateral flow immunoassay based on biotin–avidin system as a control with the sensitivity of 94.6% and specificity of 100%, targeting N protein = specific antibodies as well. Due to its sensitivity and time efficiency [13] in detection, the N protein became more attractive for diagnostic applications [14,15,16].

The immunochromatography assay was based on high specificity of antigen–antibody interaction to capture specific molecules isolated by lateral flow [17,18]. Compared with conventional methods, this technique can provide various advantages such as simplified procedure and rapid operations at low cost; it also presents immediate results without requiring skilled technicians or expensive equipment. Quantum dots (QDs), as compelling semiconductor nanocrystal fluorophores, have been observed in recent years due to their unique optical property of obtaining high quantum yields, broad absorbance peaks, narrow symmetrical emission peaks, stability against photobleaching and high signal-to-noise ratio [19,20]. Moreover, many modification strategies, typified by core–shell structures [21,22] and inorganic carrier-based multilayer nanobeads, were used to improve the fluorescent properties of QDs [23,24]. A number of immunochromatography assays based on QDs labeling were developed, targeting multiplex biological macromolecules [25], disease-associated genes [26] and toxins [27,28]. Therefore, QDs have displayed potential value for applications in cellular labeling, deep-tissue imaging and especially assay labeling as efficient fluorescence resonance energy transfer donors in molecular biological fields [29].

In this study, a fluorescent immunochromatography assay based on quantum dot nanoparticles (QDs-FICA) for point-of-care (POC) detection of SARS-CoV-2 specific antibodies was developed and evaluated, and was proved to be highly sensitive and specific. The assay provided an efficient tool for the serological diagnosis of SARS-CoV-2 infection applicable to broad mammalian species.

## 2. Results

### 2.1. Expression and Purification of Recombinant SARS-CoV-2 N Protein

As shown in Figure 1a, the recombinant SARS-CoV-2 N protein with His-Tag was expressed in *Escherichia coli* (*E.coli*) and purified efficiently by Ni-NTA affinity chromatography with a concentration of 0.762 mg/mL and a purity of 90.53%. The interaction with mouse anti-His-Tag monoclonal antibodies was verified by Western blot (Figure 1b), which indicated that biological activity was retained.

### 2.2. Characterization of the QDs-Conjugated N Protein and Optimization of Conjugating Conditions

In view of the principle of electrophoresis separation, the migration rate of N protein-modified QDs was much lower than that of QDs (Figure 1c). The peaks of fluorescence spectra showed that the fluorescence intensity of QDs decreased after modification (Figure 1d). The diameter distribution of these nanoparticles was measured by dynamic light scattering (DLS). The results of DLS showed that the average hydrated diameter of the QDs and QD-conjugated N protein were 24.38 nm and 132.20 nm, respectively (Figure 1e,f), and the ζ potential switched from −1.93 mV to −15.40 mV (Figure 1g,h). All the data above suggested efficient conjugation between QDs and N protein.

As for conjugating conditions, the ratios of EDC, SPA and pH value were optimized (Table 1, Figure 2a,b). According to the fluorescence intensity of QDs-FICA and peak values of fluorescence spectra, the most appropriate combination was as follows: an EDC ratio of 1:2500; an N protein ratio of 1:7.5; a pH of 8.0.

### 2.3. Optimization and Evaluation of QDs-FICA for Rapid Detection of Anti-SARS-CoV-2 N Antibodies

The lateral flow immunoassay was loaded onto the test strip assembled as the schematic illustration displayed (Figure 3). Briefly, 1 μL of serum was 1:100 diluted by 99 μL phosphate buffer (PBS, 50 mM and pH 7.4) in microplate well. After 3–5 times mixing, 90 μL diluted sample was absorbed by the sample pad, migrating along the strip by capillarity, and fluorescent results were displayed and interpreted. As shown in Figure 4, the optimized concentrations of coated mAb and SPA were, respectively, 0.3 mg/mL and 0.1 mg/mL detected on the control line (C line), while the optimal concentrations of mAb and SPA were both 0.3 mg/mL detected on the test line (T line) (Figure 4a,b). In order to achieve the best performance of QDs-FICA, the optimal concentration was determined to be 0.3 mg/mL. As for the buffer solution, BBS (pH 8.0) was determined to be a suitable dilution buffer (Figure 4c,d) and the best incubation time to achieve fluorescence stability for detecting mAb 2E2, polyclonal antiserum and negative control was 20 min (Figure 4e–g).

### 2.4. The Limit of Detection and Cross-Reactivity of QDs-FICA

The limit of detection of QDs-FICA was determined by a monoclonal antibody, which was fourfold diluted from 1:1 × 10^2^ to 1:1.638 × 10^6^ (with the concentration of 5 × 10^−2^ to 3.05 × 10^−6^ mg/mL). Results indicated that the limit of detection was 1:1.024 × 10^5^ with an IgG concentration of 48.84 ng ng/mL (Figure 5a,b). Cross-reactivity with positive samples (*n* = 5) containing antibodies to other relevant zoonotic coronaviruses and respiratory infection-related viruses (PRRSV, PEDV, TGEV, PRV and CSFV) was detected. The result indicated that (Figure 5c,d) the QDs-FICA performed no cross-reactivity with other coronaviruses, except for the weak interaction with PEDV-positive serum.

## 3. Discussion

From early January 2020 to present, the outbreak of pneumonia caused by SARS-CoV-2 has induced great damage to public health globally. Since it usually takes 3~8 days for symptom onset and specific antibodies appearance, rapid and accurate early diagnosis is necessary for suspected case confirmation and disease control.

To date, many immunological diagnostic approaches were developed for the detection of SARS-CoV-2-specific antibodies [3,4,5], which included enzyme-linked immunosorbent assay (ELISA) [6,7], chemiluminescent immunoassay (CLIA) [8,9], lateral flow immunoassay [10,11,12,13,14], paper-based electrochemical immunosensor [15] and microfluidic-based electrochemical immunosensor [16] using indirect [17] or sandwich [18] methods. However, a series of immunosorbent methods require a long incubation time, skilled operator and supporting instruments, rendering them difficult to promote in practical on-site testing. Due to the advantages of low costs and convenience, immunochromatographic test strips are applied in multiple fields over the past decade, such as allergies [19,20], infectious diseases [21,22], environmental contaminants [23] and drugs [24,25]. All the above render lateral flow assay one of the most efficient immunological diagnostic methods to realize point-of-care diagnosis and detect previous or ongoing infections. The immunoassay developed in this study was capable of presenting results in 15~20 min under UV light, with no professional operators or expensive equipment required, which can be a convenient approach for large-scale on-site detection.

Among SARS-CoV-2 targets of concern, the spike (S) protein has been most widely used for developing diagnostic reagents. S protein is a glycosylated trimeric class Ⅰ fusion protein engaging with indispensable functions relating to virus entry and antibody-mediated neutralization [26,27]. Three positively selected sites (S: 439N, S: 483V and S: 493Q) located in the receptor-binding domain (RBD) of spike protein are critical active sites inducing neutralization [28]. Interestingly, further studies of antibody dynamics in patients infected with SARS-CoV-2 reported that, along with antibodies to spike protein, antibodies against N protein were also generally detectable. Nucleocapsid protein is considered a highly immunogenic target that induces strong humoral responses [29,30]. Due to high genetic conservation [31] and abundant expression during virus life cycles, antibodies against N protein are appreciable targets for serological diagnosis. The fluorescent immunochromatography assay developed in this study was based on QDs-modified N protein as probes and SPA as capture molecules, targeting the Fc segment of SARS-CoV-2-specific antibodies. The complex of specific antibodies and QDs-conjugated N protein could be captured by SPA coated on a test line to present fluorescence. By contrast, non-specific antibodies were captured by the surplus binding sites of SPA without fluorescence appearing. Based on the broad species of SPA-captured antibodies, the fluorescent immunochromatography assay developed in this study overcame species limitation, fitting wider application scenarios of detection.

During conjugating, EDC served as a linking agent, which could activate carboxyl groups on the surface of QDs to form amido bonds with N proteins. Then, agarose gel electrophoresis, DLS and zeta potential were used for measuring particle size, dispersity and surface charge. The coupled probes migrated slower than bare quantum dots in electrophoresis and the major peak of hydrodynamic size indicated the contribution to nanoparticle enlargement by QDs-modification. The zeta potential was calculated using the electrophoretic mobility obtained from electrophoretic light scattering measurements [32], which could predict colloid stability and surface morphology [33]. The results have shown that the coupling affected the surface charge distribution of nanoparticles; however, destabilization, coagulation and settlement probably happened to water-dispersed nanoparticles [34], which could be influenced by pH, iron composition, ionic strength and so on [35]. On the other hand, an appropriate amount of interaction with the target protein helped enhance stability and the photoluminescence quantum yield of QDs and prevented QDs from aggregating [36]. After a series of optimizations, the most appropriate conditions for coupling were determined by using a fluorescent immunochromatography assay within a singular test line and fluorescent spectrum scanning. The optimum condition of coupling achieved the most probes synthesized so that the fluorescent performance was the best under a certain concentration of coated SPA and testing antibodies. On the contrary, QDs coagulation could be generally observed in low pH values (groups 3, 6 and 9), resulting in a decreased peak value of fluorescence spectrum scanning. Meanwhile, an improper ratio of EDC and the target protein made the conjugates inhomogeneous and prone to settle.

Among animal-derived coronaviruses, PEDV and TGEV were swine coronaviruses which were 65.21% and 63.85% homologous with SARS-CoV-2 (Appendix A). PRRSV, PRV and CSFV were respiratory infection-related viruses that presented similar symptoms (fever, emesis or diarrhea) with SARS-CoV-2. Hence, we selected the viruses above for conducting cross-reactivity detections of the developed QDs-FICA assay. During evaluation, the positive serum of PEDV presented weak fluorescence signals. The identity between the nucleocapsid protein of PEDV and SARS-CoV-2 was 44.30%, according to sequence alignment by BLAST, which was reasonable for weak cross-reactivity. The fluorescent identity of interaction with PRV was equal with the cut-off concentration (1:1.024 × 10^5^) for naked-eye observation, which was considered to be negative. Several studies reported the susceptibility of domestic pigs [37,38,39], but conclusive evidence for swine-isolated SARS-CoV-2 variants remained insufficient at this point. Since the intermediate host of SARS-CoV-2 was not quite clear to date, many efforts in cellular and molecular degrees were devoted to screening the interaction between animal counterparts of human ACE2 and SARS-CoV-2 RBD [40]. Wu et al. [41] found that SARS-CoV-2 RBD could interact with ACE2s orthologs from species belonging to *Primates*, *Lagomorpha*, *Pholidota*, *Carnivora*, *Perissodactyla*, *Artiodactyla* and *Chiroptera*. Meanwhile, many isolates [42,43,44], RT-qPCR-positive [45,46] and seropositive cases [47,48] from bats, minks, ferrets, tigers, monkeys, cats and dogs were found, indicating continuous adaptive mutations during cross-species transmission. It is conceivable that the interaction with the receptor could be enhanced, which is greatly beneficial to virus variants circulating among the host population and spilling over to humans [49]. Therefore, feasible assays and products are indispensable for serological screening and monitoring potential susceptible hosts. Sequence alignments (Jalview Software, University of Dundee, Dundee, Scotland, UK; BLAST, National Institutes of Health, MD, USA) demonstrated the high conservation (identity of 98.6~99.8%) of N proteins among SARS-CoV-2 isolates originating from domestic and wild animals in recent years (Appendix A).

## 4. Materials and Methods

### 4.1. Materials

#### 4.1.1. Plasmids, Sera and Cells

The plasmid *pET-28a-N* coding SARS-CoV-2 nucleocapsid protein, *Escherichia coli* (*E.coli*) BL21(DE3) and monoclonal antibodies (mAbs) 4B8 and 2E2 were preserved by the Provincial Key Laboratory of Biological Immunology (Zhengzhou, China). Rabbit polyclonal antiserum containing N protein-specific polyclonal antibodies and positive serum samples containing specific antibodies against PRRSV, PEDV, TGEV, PRV and CSFV were provided by Key Laboratory of Animal Immunology of Henan Academy of Agricultural Sciences (Zhengzhou, China).

#### 4.1.2. Reagents

Orange–red (*λ_em_* = 610 nm, particle size = 24 nm) ZnCdSe/ZnS QDs were purchased from Wuhan Jiayuan Quantum Dots Corporation, Ltd. (Wuhan, China). 1-(3-dimethylaminopropyl)-3-ethylcar-bodiimide hydrochloride (EDC-HCl) and Bovine serum albumin (BSA) were purchased from Sigma-Aldrich Chemical Co. (St. Louis, MO, USA). BCA Protein Assay Kits (PC0020) were purchased from Sorlarbio Life Sciences (Beijing, China). *Staphylococcal* protein A (SPA) was purchased from Beijing Bersee Science and Technology Corporation, Ltd. (Beijing, China). HRP-conjugated anti-His-Tag monoclonal antibody (5C3) and HRP-conjugated goat anti-mouse IgG were purchased from Abbkine Scientific Corporation, Ltd. (Wuhan, China). eECL Western Blot Kit was purchased from CWbiotech Corporation, Ltd. (Jiangsu, China). Nitrocellulose membranes, glass fiber and absorbent pads were purchased from Millipore (Bedford, MA, USA).

### 4.2. Expression and Purification of Recombinant SARS-CoV-2 Nucleocapsid (N) Protein

Recombinant SARS-CoV-2 N protein with histidine tag at N-terminus was expressed in *Escherichia coli*. and purified via Ni-NTA affinity chromatography using HisTrap^TM^ excel (GE, 17-3712-06, Boston, MA, USA). An aliquot of 2 mL of overnight *E.coli* BL21(DE3) culture was expanded into 200 mL of LB medium containing 10 μg/mL Kanamycin and incubated at 37 °C for 1.5 h, and 0.1 mmol/L IPTG was used to induce the expression of N protein for 12 h at 16 °C. The cell pellets were harvested and sonicated on ice for 15 min. The turbid solution was then centrifuged for 15 min at 12,000 g, and the supernatant was collected for purification.

During affinity chromatography, the concentration of each tube was monitored with ultraviolet absorption at 280 nm and the eluted fraction of target protein was collected successively. Saline ion reduction was achieved by dialysis in 50 mM Tris-HCl (pH 8.0) solution. Purified N proteins were analyzed by 12% SDS-PAGE (Mini-Protean and Bio-Rad Gel Doc XR Imager, Biorad, Hercules, CA, USA) and Western blot. N protein separated by SDS-PAGE was briefly transferred to methanol-activated PVDF membrane. The membrane was blocked with 5% skimmed milk at 4 °C overnight and then incubated with HRP-conjugated anti-His-Tag monoclonal antibodies (diluted at 1:5000) for 1 h at 37℃. After vibratory washing with PBST (phosphate buffered saline with 0.5% Tween-20) three times, the target band was exposed by electrochemiluminescence (ECL) with eECL Western Blot Kit. The purity of N protein was measured with Image J (Version 1.8.0, National Institutes of Health, MD, USA), and the concentration was determined at the absorbance of 562 nm using the BCA Protein Assay Kit and bovine serum albumin as standard.

### 4.3. Preparation of QDs-Conjugated N Protein and Optimization of Synthesis Procedure

#### 4.3.1. Preliminary Synthesis of QDs-Conjugated N Protein

By means of the formation of amide bond (CO-NH), conjugates were prepared by immobilizing N protein on the surface of QDs. In detail, 2.5 μL of QDs was added into 7.66 μL EDC-HCl (1 mg/mL) to activate carboxyl groups and the mixture was incubated in the dark for 30 min at a temperature of 25 °C and a shaking speed of 220 rpm. Then, 20 μL of nucleocapsid (N) protein and 19.84 μL of phosphate buffer (PBS, 50 mM and pH 8.0) were added to the mixture and incubated on the shaking incubator for another 3 h with the same conditions. To block reductant active carboxyl groups, 1% BSA was used and the conjugates were stored at 4 °C and identified using agarose gel electrophoresis. The fluorescence spectra of QDs and QDs-conjugated nanoparticles were both analyzed using SpectraMax i3x Platform (Molecular Devices, Shanghai, China). The dispersity and diameters of nanoparticles above were characterized by Dynamic Light Scattering (DLS) with Malvern Zetasizer (Malvern Panalytical Ltd., Malvern, UK).

#### 4.3.2. Optimization of Conjugating Conditions

The inputs of carboxyl-active agent, N protein and pH value were considered effective factors during conjugation; hence, a series of conditions such as the ratio of EDC, the ratio of N protein and pH was optimized by an orthogonal experiment described in Table 1. The optimal group of conjugating conditions was selected according to the peak value of the fluorescence spectrum curves and the fluorescence intensity performed on the test strip, which was recorded by an MD-980 Multi-channel Fluorescent Immunoassay Analyzer (Microdetection Corporation, Ltd., Nanjing, China).

### 4.4. Preparation, Optimization and Evaluation of Immunochromatographic Test Strip for Rapid Detection of Antibodies against SARS-CoV-2 N Protein

#### 4.4.1. Assembly of Immunochromatographic Test Strip and Interpretation of the Results

A lateral flow immunoassay for rapid detection of specific antibodies against SARS-CoV-2 N protein was loaded on the test strip, which was assembled as follows: sample pad (300 mm × 15 mm), conjugate pad (300 mm × 8 mm), absorbent pad (300 mm × 18 mm), and NC membrane (300 mm × 15 mm) attached to a semi-rigid polyethylene sheet. Specifically, sample pads were soaked with treatment fluid (0.1 M PBS, containing 5% sucrose, 0.05% sodium azide, 0.3% Tween 20 and 1% BSA) and dried at 42 °C for 4 h. The conjugate pad, test line (T line) and control line (C line) were, respectively, coated with QDs-labeled N protein, *Staphylococcus aureus* protein A (SPA) and mAb 4B8 (with the titer of 1:2.048 × 10^6^ and the *K*_aff_ of 1.22 × 10^9^ L/mol, Appendix A), using XYZ Biostrip Dispenser (Biodot, Richmond, CA, USA). After being assembled with all sections, the sheet was further cut into strips (50 mm × 3 mm) by CM 4000 Cutter (Biodot, Richmond, CA, USA) and packaged in a plastic bag with silica desiccant. The schematic diagram of overall structure of the test strip is shown in Figure 3a.

The serum sample was pipetted and added to the microwell plate with a dilution buffer (PBS, 50 mM and pH 8.0). After being dropped in the sample well of the test strip, it would redissolve the QDs-labeled N protein on the conjugate pad. Through capillary action, the diluted sample migrated to the absorbent pad. If the specimen contains specific antibodies against the SARS-CoV-2 N protein, the antigen-antibody complex will be captured by SPA on the test line and redundant QDs-N probes will be captured by mAbs on the C line resulting in a positive antibodies reading. If no N protein antibodies are present, a negative result will be displayed so that only the C line is visible. If the C line cannot be observed, the strip is considered invalid and another performance is required for the test. The entire test is supposed to be visible under a handheld UV light (Crystal Technology & Industries, Inc., Dallas, TX, USA) within 15 min (Figure 3b).

#### 4.4.2. Optimization of Concentration of Coated Antigens, Antibodies and Experimental Conditions

Due to the possibility that the fluorescent intensity of the test strip could be influenced by hook effect [50,51], suitable concentrations of coated antigens and antibodies were determined for the purpose of proper immunological reactions and best sensitivity. SPA coated on the T line and mAb 4B8 on the C line were both set to be 0.3, 0.5 and 0.7 mg/mL. A series of test strips was used to detect mAb 2E2 1:10,000 (with the titer of 1:3.2 × 10^4^ and *K*_aff_ of 2.5 × 10^9^ L/mol (Appendix A)) by the Fluorescent Immunoassay Analyzer.

Experimental conditions including different dilution buffers (PBS, BBS and TBS) with different pH values (6.2, 6.8, 7.4, 8.0 and 8.6) and reaction time (5 min, 10 min, 15 min, 20 min, 25 min and 30 min) were further optimized to improve detection performance.

#### 4.4.3. Limit of Detection, Cross-Reactivity and Reproductivity of the QDs-FICA

In order to verify the limit of detection (LOD), mAb 2E2 against N protein were fourfold serially diluted from 1:1 × 10^2^ to 1:1.638 × 10^6^ (with the concentration of 5 × 10^−2^ to 3.05 × 10^−6^ mg/mL) by negative serum and tested in triplicate. The results were observed within 10 min, and the lowest antibody concentration presenting color reaction on the corresponding test line was the limit of detection (LOD) of the test strip. The control line should always appear if the test procedure is performed orderly and the reagents are working as intended.

Cross-reactivity was tested using positive serum samples (*n* = 5), which contain antibodies against 2 strains belonging to Coronaviridae and 3 strains of viruses causing respiratory infection to evaluate the specificity of the lateral flow immunoassay.

## 5. Conclusions

In summary, a universal fluorescent immunochromatography assay based on quantum dot nanoparticles (QDs-FICA) for the point-of-care detection of antibodies against SARS-CoV-2 N protein was developed and evaluated in this study. The limit of detection was determined to be 1:1.024 × 10^5^ with an IgG concentration of 48.84 ng/mL, and good specificity was found by detecting five strains of relative zoonotic viruses and respiratory infection-related viruses. The QDs-based lateral flow assay provides a reliable method for rapid and sensitive detection and large-scale serological screening of SARS-CoV-2 infection applicable to broad mammalian species, which can be conducive to understanding and monitoring host migration of other viruses transmitting from animals to humans.

## Figures and Tables

**Figure 1 ijms-23-06225-f001:**
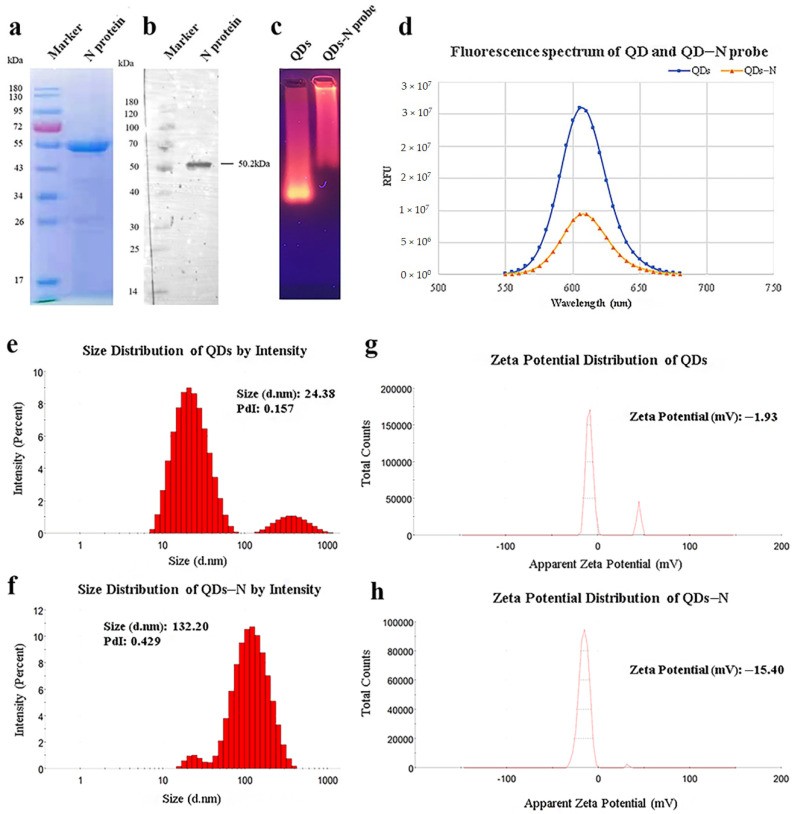
Identification of purified nucleocapsid protein and QDs-conjugated N protein. (**a**) Purified nucleocapsid protein isolated by 12% SDS-PAGE. (**b**) Purified nucleocapsid protein isolated by Western blot. Lane 1: Protein Ladder. Lane 2: N protein reacted with anti-His-Tag mouse monoclonal antibody (5C3). (**c**) QDs and conjugates identified by agarose gel electrophoresis. (**d**) Fluorescence spectrum of QDs and conjugates. (**e**–**h**) Size distribution and ζ potential distribution of QDs and conjugates, as measured by Malvern Zetasizer.

**Figure 2 ijms-23-06225-f002:**
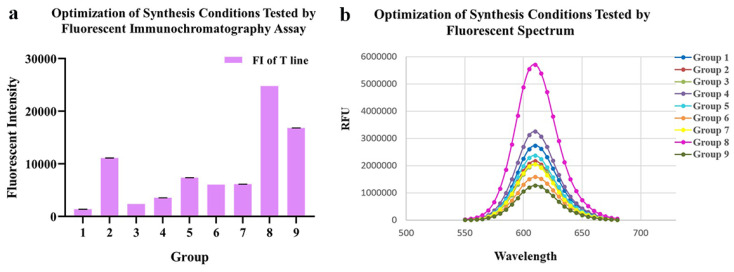
Optimization of conjugating conditions of QDs-N probes. (**a**) Fluorescent intensity of 9 probes synthesized with different conditions tested by fluoro-immunoassay. (**b**) Fluorescent Spectrum of 9 probes, which indicated that the conjugation 8 presented the best fluorescent performance.

**Figure 3 ijms-23-06225-f003:**
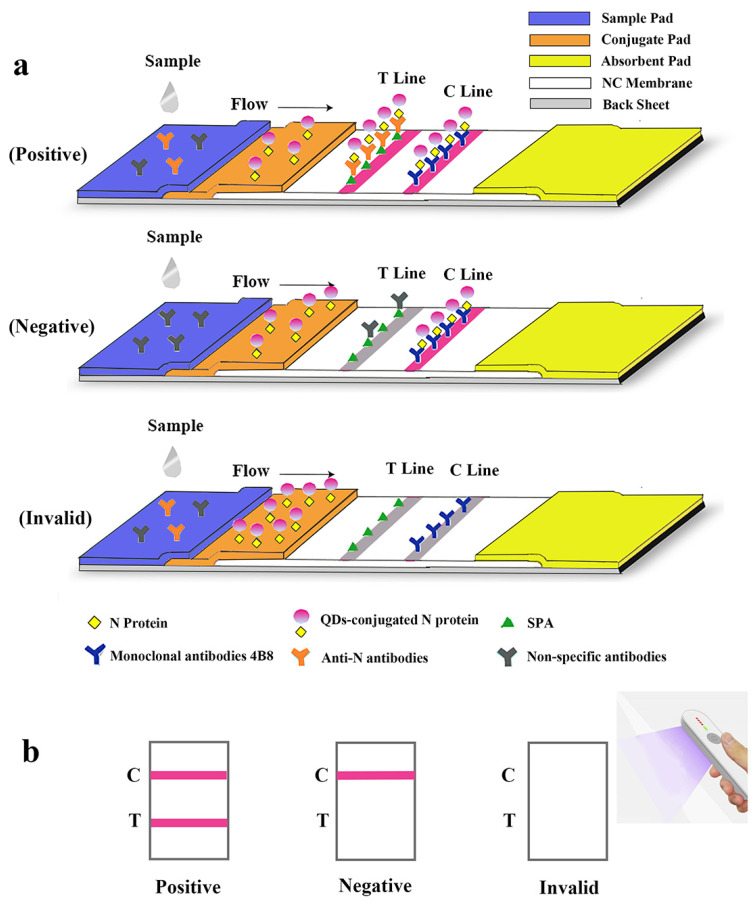
Schematic illustration of the lateral flow assay for the rapid detection of antibodies against SARS-CoV-2 N protein. (**a**) Schematic diagram of the detection device. (**b**) Interpretation of different testing results under UV light. “C” and “T” refer to the control line and the test line, respectively.

**Figure 4 ijms-23-06225-f004:**
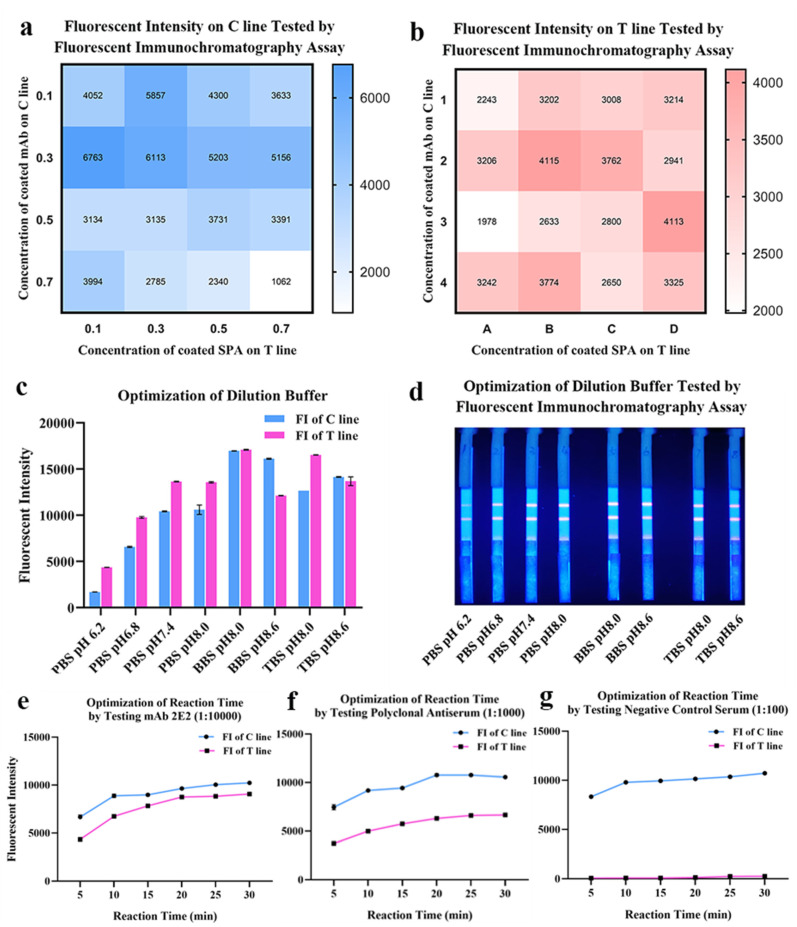
Optimization of coating concentrations, pH and buffer solution of the lateral flow assay. (**a**,**b**) Optimization of coated mAbs and SPA on the T line and the C line. The fluorescent intensity data was presented by heat map. (**c**,**d**) Optimization of the dilution buffer and pH. (**e**–**g**) Optimization of reaction time by testing mAbs, antiserum and negative control.

**Figure 5 ijms-23-06225-f005:**
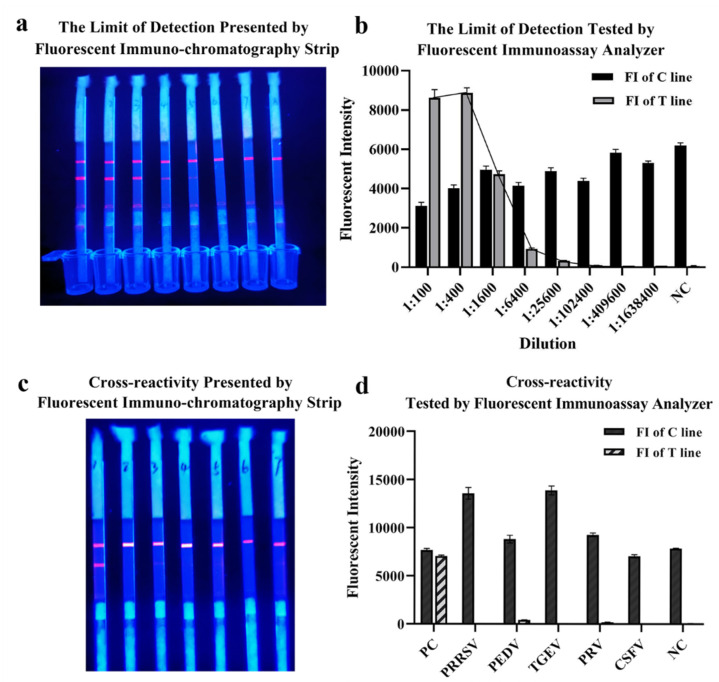
The limit of detection and cross-reactivity of the lateral flow assay. (**a**,**b**) The limit of detection of the lateral flow assay. (**c**,**d**) Cross-reactivity with other coronaviruses and respiratory-infection-related viruses (*n* = 5).

**Table 1 ijms-23-06225-t001:** Optimization of conjugating conditions of QDs-N probe by using orthogonal tests.

Group	The Ratio of EDC	The Ratio of N Protein	pH of PBS	FI of Test Line
1	1:1500	1:2.5	7.4	1372 ± 24
2	1:2000	1:2.5	8.0	11,109 ± 22
3	1:2500	1:2.5	6.8	2368 ± 4
4	1:2500	1:5.0	7.4	3551 ± 10
5	1:1500	1:5.0	8.0	7375 ± 3
6	1:2000	1:5.0	6.8	6050 ± 4
7	1:2000	1:7.5	7.4	6145 ± 8
8	1:2500	1:7.5	8.0	24,797 ± 5
9	1:1500	1:7.5	6.8	16,817 ± 14

## Data Availability

The study did not report any public archived data.

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
