# Peer review of "A Universal Fluorescent Immunochromatography Assay Based on Quantum Dot Nanoparticles for the Rapid Detection of Specific Antibodies against SARS-CoV-2 Nucleocapsid Protein"

_ijms, 2022, doi:10.3390/ijms23116225_

Round 1
Reviewer 1 Report
Authors, Li et al present here an immunochromatography assay for rapid detection of SARS-CoV-2 N protein.
Major comments:
1. No information on how the serum sample were obtained – positive, cross-reactive, etc. Please provide information on relevant ethical clearance as need.
Minor comments:
1. Line 12. What do authors mean by ‘main’?
2. Line 15: conservative among variants is not true. Please delete
3. Throughout the manuscript, the authors describe the test and control lines as C and T lines, although they have not been defined in the manuscript (except in figure 3). Please mention it in Line 306.
4. Line 116-117: It would be helpful to find the information on sample volume and running (dilution) buffer volume in the methods.
5. Section 4.1: Please mention relevant information on sample pad; conjugate pad; NC; wicking pad etc
6. Line 350-351: Authors mention no cross-reactivity was seen. However, in Figure 5D: There is some signal in both PEDV and PRV. Please explain and include the observation in the result and discussion
1. 6: Figure 6a and 6b: Is not needed to just make a statement regarding homology. Please remove.
7. Line 305: elaborate on the composition of ‘treatment fluid’.
8. Line 312: elaborate on the composition of ‘dilution buffer’
Author Response
Dear reviewer:
Thank you for your valuable comments to this work. We provided response in turn to each question in the following Word file. Please see the attachment.
Wish you all the best!
Sincerely yours,
Gaiping Zhang

Reviewer 2 Report
Dear author,
It is an interesting study and the manuscript is well-written.
I have some questions:
1. Could you precise if your quantum dot was designed for all SARS-CoV-2 variant?
2. Could this technique be used to detect sARS-CoV-2 in a nasopharyngeal swab? in saliva?
3. If so, what would be the advantages and disadvantages compared to PCR which is currently the universal technique?
4. Have you tested your quantum dot on patient samples?
Author Response
Dear reviewer:
Thank you so much for your insightful comments. We provided response to each question in the following Word file. Please see the attachment.
Wish you all the best!
Sincerely yours,
Gaiping Zhang

Reviewer 3 Report
The manuscript is well written and it could be accepted in the present form.
Author Response
Dear reviewer,
Thank you very much for your approbation, which is a great encouragement to our work. We will take efforts to contribute more research for this field.
Wish you happy and all the best.
Sincerely yours,
Gaiping Zhang